# Effects of Fiber Volume Fraction and Length on the Mechanical Properties of Milled Glass Fiber/Polyurea Composites

**DOI:** 10.3390/polym14153080

**Published:** 2022-07-29

**Authors:** Jing Qiao, Quan Zhang, Chong Wu, Gaohui Wu, Longqiu Li

**Affiliations:** 1School of Materials Science and Engineering, Harbin Institute of Technology, Harbin 150001, China; 21b909055@stu.hit.edu.cn (Q.Z.); zjbnylwl@163.com (C.W.); wugh@hit.edu.cn (G.W.); 2School of Mechatronics Engineering, Harbin Institute of Technology, Harbin 150001, China

**Keywords:** polyurea elastomer, milled glass fiber, polymer matrix composites, mechanical properties

## Abstract

Composites of polyurea (PU) reinforced with milled glass fiber (MG_f_) were fabricated. The volume fraction and length of the milled glass fiber were varied to study their effects on the morphological and mechanical properties of the MG_f_/PU composites. The morphological attributes were characterized with scanning electron microscopy (SEM) and Fourier transform infrared (FTIR) spectroscopy. The SEM investigations revealed a uniform distribution and arbitrary orientation of milled glass fiber in the polyurea matrix. Moreover, it seems that the composites with longer fiber exhibit better interfacial bonding. It was found from the FTIR studies that the incorporation of milled glass fiber into polyurea leads to more phase mixing and decreases the hydrogen bonding of the polyurea matrix, while having a negligible effect on the H-bond strength. The compression tests at different strain rates (0.001, 0.01, 0.1, 1, 2000 and 3000 s^−1^) and dynamic mechanical properties over the temperature range from −30 to 100 °C at 1 Hz were performed. Experimental results show that the compressive behavior of MG_f_/PU composites is nonlinear and strain-rate-dependent. Both elastic modulus and flow stress at any given strain increased with strain rate. The composites with higher fiber volume fraction and longer fiber length are more sensitive to strain rate. Furthermore, the elastic modulus, stress at 65% strain and energy absorption capability were studied, taking into account both the effect of fiber volume fraction and mean fiber length. It is noted that an increase in fiber volume fraction and fiber length leads to an increase in elastic modulus, stress at 65% strain and absorbed energy up to ~103%, 83.0% and 137.5%, respectively. The storage and loss moduli of the composites also increase with fiber volume fraction and fiber length. It can be concluded that the addition of milled glass fiber into polyurea not only improves the stiffness of the composites but also increases their energy dissipative capability.

## 1. Introduction

Polyurea elastomer is a promising protective coating against the mitigation of blast and ballistic impact [1,2,3,4,5,6,7], owing to its light weight, high elongation and strength, excellent viscoelasticity and impact resistance. Moreover, it exhibits very good adhesion and can be employed to improve the dynamic performance of a variety of substrates—metallic plates [8,9,10,11], concrete [12,13,14,15] and other composite structures [16,17,18,19]. In order to further improve the mechanical properties of polyurea, the addition of various substances into polyurea to prepare composites has been demonstrated to be an effective approach. For example, Toader et al. [20] integrated multi-walled carbon nanotubes (MWCNTs) into polyurea, and improved impact load resistance was obtained. Their results revealed that a 0.2% addition of MWCNTs could increase the tensile strength and deformation energy of the composite by 27% and 56.4%, respectively. Liu et al. [21] developed SiC/polyurea nanocomposites and found that 0.7 wt.% SiC/polyurea composites had the highest strain rate sensitivity, mechanical properties and strain energy, implying a clear advantage in resistance to shock loads. Barczewski et al. [22] demonstrated that the tensile strength, elastic modulus and elongation at the breaking of the 20 wt.% basalt powder/polyurea composite were increased by 40.4%, 72.9% and 99.7%, respectively.

Although incorporation of nanoparticles in polyurea leads to enhanced mechanical properties, difficulty in uniform dispersion and the expensive price of nanoparticles limit their applications. Furthermore, fiber-reinforced composites generally exhibit higher toughness and impact loss tolerance compared to particle-reinforced composites due to energy dissipation mechanisms such as fiber deformation, crack bridging, interfacial debonding and fiber pullout [23,24]. Song et al. [25] studied the strengthening effect of glass-fiber-reinforced polyurea composites on concrete structures, and the results showed that concrete structures sprayed with glass fiber/polyurea composites had better bearing capacity and flexural ductility than pure polyurea. Carey [26] fabricated short E-glass-fiber-reinforced polyurea composites and researched their strength and ductility by tensile testing. It was elucidated that the strength of their fiber-reinforced polyurea composites increased as the fiber content and the stress at 5% elongation increased up to 400% at the addition of 11.5% fiber. These were quite promising results. However, the ductility of their composites was unsatisfactory. Elongation at the breaking of the composites was less than 20%. Furthermore, other properties such as the microstructure, physical properties and dynamic mechanical properties have not been reported. Nemat-Nasser et al. [27,28,29] developed polyurea-based composites using milled glass fiber with 200 μm length. Dynamic mechanical properties were characterized using dynamic mechanical analysis (DMA) and ultrasonic measurements. The results indicated that both storage and loss moduli of the composites increased drastically with fiber volume fraction, up to 4 times compared with the pure polyurea. Moreover, they mentioned that the composites possessed both high stiffness and high energy loss at higher temperature, which implied that the composites could be an excellent protective candidate for critical applications in ballistic and blast loading. Nevertheless, their studies mainly focused on mechanical models, and no more experimental results on such composites were reported. In our previous work [30,31], a kind of waste product of thermal power station with micron size, fly ash hollow spheres was used to develop polyurea-based composites, and the results are encouraging. The elongation at the breaking of the composites with 20 vol% fly ash particles is still larger than 300%, which is enough for most applications. At the same time, the stress at 300% elongation of the composites is 40% higher than that of the pristine polyurea matrix. Moreover, the dynamic mechanical properties of the composites with 30 vol% fly ash particles at low frequencies increased by more than 200% than those of the pristine polyurea matrix. Therefore, in the present work, we developed milled glass fiber/polyurea composites using our fabrication process with the aim of obtaining improved mechanical properties while maintaining good ductility.

It is well known that the efficiency of reinforcing composites with a fibrous filler depends on their geometrical parameters, volume fraction and orientation in the material body. For short-fiber-reinforced polymer (SFRP) composites, fiber content and fiber length are two of the most important parameters that have evident influence on their mechanical properties [32,33]. Generally, increasing the fiber content and fiber length leads to an increase in composite strength, modulus and toughness. However, the difficulty in the fabrication of SFRP composites and probability of fiber breakage increases with increased fiber length and fiber content [34,35]. Sreenivasan [35] studied the mechanical properties of short Sansevieria cylindrical fiber/polyester composites and found that there is a critical fiber length and optimum fiber weight percent. Therefore, research on the effect of fiber content and fiber length on the mechanical properties of SFRP composites is of particular interest and significance.

Therefore, in the present work, milled glass fiber/polyurea composites reinforced with various fiber content and fiber length were prepared. Their microstructures were investigated with scanning electron microscopy (SEM) and Fourier transform infrared (FTIR) spectroscopy. The stress–strain characteristics of the composites at six different strain rates were analyzed. Furthermore, dynamic mechanical analysis of the composites was conducted over the temperature range from −30 to 100 °C at 1 Hz. The effect of fiber length and volume fraction is discussed.

## 2. Materials and Methods

### 2.1. Materials

Basic materials employed in this investigation were polycarbodiimide-modified diphenylmethane diisocyanate (Isonate 143L) from Dow Chemical, poly(tetramethylene oxide-di-p-aminobenzoate) (PTMO) (Versalink P-1000) from Air Products and milled glass fiber from Hangzhou Corker Composite Material Co., Ltd. The average isocyanate (NCO) functionality of Isonate 143L provided by the manufacturer is 2.1. The molecular weight of Versalink P-1000 given by the manufacturer is 1238. Milled glass fibers with a silane-based surface treatment, diameter of 13 μm and density of 2.1 g/cm^3^ were selected in this study. The milled glass fibers were obtained by milling short E-glass fiber that had alkali content less than 0.8% and were then sieved with a standard mesh sieve column. In order to investigate the effect of fiber length on the properties of the milled glass fiber/polyurea composites, three fiber length classes (long (20 mesh), medium (50 mesh) and short (200 mesh)) were used in this work. The length distribution of the glass fiber was studied based on an image analysis method. One single layer of fibers was sprayed on a microscope slide. An Axiovert 40 MAT microscope was employed for image acquisition. Fiber length and distribution were subsequently obtained by analyzing the images using Image Tools. Figure 1 shows the typical SEM images of the milled glass fibers and their length distribution. The average fiber lengths were calculated as 154.1, 103.9 and 53.7 μm, respectively.

### 2.2. Preparation of Composites

Milled glass fibers were introduced into Versalink P-1000 in a predetermined proportion, and the mixture was stirred for 2 h under vacuum using a magnetic stirrer. Then, the previously vacuumed Isonate 143L was poured into the above blend and mixed for five minutes under vacuum. Finally, the mixture was cast into Teflon molds. The stoichiometric ratio of isocyanate (Isonate 143L) to amine (Versalink P-1000) was 1.0. Two groups of specimens were prepared. The first group was made to study the effect of fiber length on the mechanical properties of the MG_f_/PU composites. Fibers with three different lengths mentioned above were used. The fiber volume fraction was calculated as the measured weight of the fibers incorporated into the specimen divided by the fiber density and was fixed at 10%. From short to long fiber length, the composites were labeled as S10/PU, M10/PU and L10/PU. The second group was for investigating the effect of fiber volume fraction on the mechanical properties of the composites. Four different fiber volume fractions were used: 0%, 5%, 10% and 15%. The fiber length was fixed at 20 mesh, i.e., 154 μm. These composites were labeled as PU, L5/PU, L10/PU and L15/PU, respectively. Prior to measurements, the specimens were cured in a desiccator at room temperature for at least two weeks.

### 2.3. Tests

#### 2.3.1. Scanning Electron Microscopy (SEM)

In order to observe the fiber distribution in the polyurea matrix and the interfacial bonding state, a Helios Nanolab600i scanning electron microscope was used to study the brittle fracture surfaces of the composites. Composite specimens were notched by razor blade and then immersed in liquid nitrogen until thermal equilibrium was achieved, at which point they were removed and fractured immediately. Due to the poor conductivity, the fractured surfaces were coated with a thin layer of platinum. The acceleration potential was 20 KV.

#### 2.3.2. Fourier Transform Infrared (FTIR) Spectroscopy

FTIR spectroscopy was conducted using a Perkin-Elmer Spectrum One FTIR Spectrometer over a wavenumber range from 4000 to 400 cm^−1^. Diamond attenuated total reflectance cell (ATR) was used. The scanning number of each experiment was 10, and the resolution was 2 cm^−1^.

#### 2.3.3. Compression Tests

The stress–strain behavior of the material in the quasi-static regime was evaluated by constant displacement uniaxial compression tests. The testing was conducted on an Instron 5569 universal testing machine. Load accuracy was ±0.4% of the reading. The specimens were subjected to three constant displacement rates of 0.75, 7.5 and 75 mm/min. The specimens were of a cylindrical geometry with diameter of 8 mm and thickness of 12.5 mm. As a result, the strain rate values for the specimens were 0.001, 0.01 and 0.1 s^−1^, respectively. Three specimens of each material configuration were tested, and average values were reported. Moderate strain rate compression testing was performed on a Gleeble 1500D thermo-mechanical simulator. Displacement control mode and cylindrical specimens with the same dimensions as the tests at low strain rates were used. The maximum displacement rate of the equipment is 1200 mm/s. In theory, the strain rates for the specimens should be up to 100 s^−1^. However, when the strain rates were larger than 10 s^−1^ for the pure polyurea specimens and 1 s^−1^ for the composite specimens, the acquired stress–strain curves were too noisy for any useful information to be retrieved. Hence, only results at the strain rate of 1 s^−1^ are reported herein. The stress–strain response of the material at high strain rates was investigated using the split Hopkinson pressure bar (SHPB) apparatus in compression. Aluminum incident and transmission bars were employed. Cylindrical specimens with a diameter of 8 mm and a thickness of 5 mm were used. The stain rates used were 2000 and 3000 s^−1^.

#### 2.3.4. Characterization of Dynamic Mechanical Properties 

The dynamic mechanical properties were determined using a Q800 dynamic mechanical analyzer from TA Instruments. A single cantilever clamp was used. The temperature dependences of the storage and loss moduli were acquired in the temperature range from −30 to 100 °C with a heating rate of 3 °C/min at a frequency of 1 Hz. The strain amplitude was 15 μm. The specimens were 17.5 mm long × 10 mm wide × 3 mm thick.

## 3. Results and Discussion

### 3.1. Microstructure

Figure 2 shows the microstructure of the MG_f_/PU composites with varying fiber lengths and fiber volume fractions. From Figure 2a–e, it can be seen that the milled glass fiber is distributed homogenously in the polyurea matrix for all the cases. No significant sign of agglomerates is observed. Moreover, the orientation of the fiber is random, which is expected since the components were mixed using a magnetic stirrer for 2 h and poured directly into an open mold.

According to the experimental method, the observed surface is the fractured surface of the composites in the glassy state. It is clear that some fibers are projected out from the matrix, and there are also many hollow cylinders or cavities, indicating that some fibers are pulled out from the matrix during the fracture process. For the composites with longer glass fibers, the surfaces of the projecting end of the fibers are not as smooth or clear as their original states, as shown in Figure 2a–c. They are covered with a thick layer of polymer. Additionally, from Figure 2f, it can be seen that the surface of the hollow cylinders in the polyurea matrix, which were created from the pulled-out fiber, is coarse. All of these phenomena indicate that the interfacial adhesion between the fibers and the polyurea matrix is good, implying a strong mechanical performance. This may be due to the surface treatment of the glass fiber with silane coupling agent by the manufacturer. The increase in the fiber volume fraction has no considerable effect on the interfacial bonding of the composites. However, it should be noted that, as the fiber length decreases, the surface of the projecting fiber becomes smoother, implying a decreasing interfacial bonding strength.

### 3.2. Hydrogen Bonding

The local hydrogen bonding nature of the MG_f_/PU composites was investigated with ATR-FTIR spectroscopy, which is very sensitive to hydrogen bonding. Figure 3 displays the FTIR spectra of pure polyurea and MG_f_/PU composites with varying fiber volume fractions and fiber lengths in the C=O (~1740–1620 cm^−1^) and N-H stretching regions (~3460–3200 cm^−1^). The spectra were normalized using the aromatic C=C absorbance at 1594 cm^−1^ as the standard. It can be seen that the FTIR spectrum of the pure polyurea displays bands at ~1714, ~1668 and ~1644 cm^−1^, associated with free, disordered and ordered hydrogen-bonded C=O, respectively. On the contrary, characteristic N-H stretching bands are observed at ~3444, ~3352 and ~3305 cm^−1^, corresponding to free N-H, disordered and ordered hydrogen-bonded N-H. Moreover, the peak at ~3260 cm^−1^ represents the stretching vibrations of N-H bonded to ether oxygen atoms in the PTMO backbone (N-H…O). The spectra obtained match those previously reported [36,37,38]. The functional group bands of the composites are identical to those of pure polyurea, and no new peaks appear. Moreover, there are no visible changes in the characteristic peak position, indicating that the H-bond strength of polyurea was not altered by the presence of milled glass fibers. With the addition of milled glass fibers, the intensity of the free C=O band increases, and a decreasing trend is observed in disordered and ordered H-bonded C=O. The composites with varying fiber lengths possess similar band intensity, except for higher ordered H-bonded C=O band intensity of the S10/PU composites. In the N-H region, introducing milled glass fibers into polyurea also leads to a decrease in the peak intensity of the disordered and ordered H-bonded N-H bands, which also decreases with increasing fiber length. However, the free N-H groups are few, and most of the groups existed in the form of hydrogen bonds.

A hydrogen-bonding index *R* defined by [39],
(1)$$R=R_{o}+R_{dis}=\frac{A_{1644}}{A_{1714}}+\frac{A_{1668}}{A_{1714}}$$
where *R_o_* and *R_dis_* are the ordered H-bonding and disordered H-bonding, respectively, and *A*_1644_, *A*_1668_ and *A*_1714_ are the intensities of the characteristic peaks at 1644, 1668 and 1714 cm^−1^ that can be used as a measure of the hydrogen bonding degree for urea carbonyl groups. The degree of phase separation (*DPS*) and the degree of phase mixing (*DPM*) can be obtained by [40],
(2)$$DPS=\frac{R}{R+1}$$
(3)DPM=1−DPS

The obtained values of *R*, *DPS* and *DPM* of pure polyurea and MG_f_/PU composites are given in Table 1. It can be seen that the H-bonding degree for urea carbonyl groups and the resultant *DPS* decreases with increasing fiber volume fraction. This is consistent with the observation of Tien et al. [40] on montmorillonite/polyurethane nanocomposites. Fiber length also affects the hydrogen bonded environment significantly. Both *R* and *DPS* decrease monotonically with increasing fiber length. This means that the addition of milled glass fiber leads to more phase mixing and decreases the hydrogen bonding.

### 3.3. Compression Testing: Rate Dependence of Stress–Strain Behavior

#### 3.3.1. Stress–Strain Relationship

Previous studies [41,42,43] have shown that the compressive stress–strain behavior of pure polyurea is highly nonlinear and strongly rate-dependent. A similar phenomenon has been observed for MG_f_/PU composites as shown in Figure 4. Due to the excellent ductility of the composites, fractures did not occur during the tests. As a result, all the tests were interrupted when strain levels reached ~70% at strain rates in the range from 0.001 to 1 s^−1^. Figure 4a gives the stress versus strain curves of L10/PU composites with different strain rates. Nonlinearities and rate dependences are clear. The stress value for a given strain level was found to increase with strain rate. Moreover, the whole stress–strain curve can be divided into three regions: an initial linear elastic region, a plateau region and a terminal nonlinear increasing region before rupture. It can be seen that the higher the strain rate, the more obvious the linear feature that emerges in the initial stage. Figure 4b,c shows the stress versus strain curves of MG_f_/PU composites with various fiber volume fraction and fiber length at the strain rate of 0.001 and ~3000 s^−1^, respectively. It can be found that the material parameters have a significant impact on the stress versus strain response. The curves increase gradually with increasing fiber volume fraction and fiber length, implying the strengthening effect of the glass fiber.

#### 3.3.2. Compressive Mechanical Properties

The influence of fiber length and volume fraction on the compressive mechanical properties of MG_f_/PU composites such as Young’s modulus, stress at 65% strain and absorbed energy at strain rates in the range from 0.001 to 1 s^−1^ is shown in Figure 5. 

Least squares analysis up to the proportional limit was used to determine the Young’s modulus of MG_f_/PU composites. Its dependences on fiber volume fraction and fiber length at strain rates 0.001, 0.01 and 0.1 s^−1^ are shown in Figure 5a,b. From Figure 5a, it can be seen that the Young’s modulus of the composites tends to increase linearly as fiber volume fraction increases for any given strain rate level. It is worth noting that strain rate also shows a significant effect on such dependence. It seems that the Young’s modulus of the composites is less sensitive to the fiber volume fraction at lower strain rates. For example, the Young’s modulus of L15/PU composites is ~86% higher than that of pure polyurea at the lowest rate (0.001 s^−1^), while the increment is ~103% at higher strain rates (0.1 s^−1^). Moreover, similar to many polymer-based composites, the modulus of the composites increases with the strain rate. In addition, it can be easily found that the addition of glass fiber effectively enhances the strain rate sensitivity of the material, e.g., compared with the value at the lowest rate (0.001 s^−1^), the increases of pure polyurea at higher strain rates (0.01, 0.1 s^−1^) are 4% and 12%, respectively, while those of L15/PU composites are 13% and 23%. This observation is consistent with our previous results on fly ash/polyurea composites. In this study, we also demonstrate that the reinforcing effect of glass fiber is dependent on the fiber length. Figure 5b shows the variations of the Young’s modulus as a function of fiber length for MG_f_/PU composites. It can be observed that the effect of fiber length is similar to that of fiber content. The Young’s modulus of the composites and their strain rate sensitivity also increase with the fiber length in the region under study, while the effect is not as significant as that of fiber content. 

Since fracturing of the composites did not occur during the compression tests, which were interrupted when strain levels reached ~70%, the stress at 65% strain was chosen to evaluate the effects of fiber volume fraction and length on the performance of the composites with large deformation. The compressive stress at 65% strain of various composites is given in Figure 5c,d. Moreover, the absorbed energy which is identical to the area under the stress versus strain curve was calculated by integrating the curves numerically up to the point with maximum strain, as shown in Figure 5e,f. From the figures, it is clear that the addition of milled glass fiber improves the stress and the energy absorption capacity of the composites. For the effects of fiber volume fraction and fiber length, the data follow the same trend as observed for the Young’s modulus of the composites and emphasize the fact that the stress and energy absorption capacity of the composites are sensitive to the fiber volume fraction, fiber length and strain rate as well. As the fiber volume fraction, fiber length and strain rate increase, the stress and energy absorption capacity of the composites are remarkably increased. When the fiber volume fraction increases to 15%, the stress at 65% strain and absorbed energy of the composites increases up to 83% and 137.5%, respectively, compared to those of pure polyurea. At high strain rate, the effect of fiber volume fraction is more significant. Similarly, the composites with higher fiber volume fraction are more strain-rate-sensitive. As the strain rate increases from 0.001 to 1 s^−1^, the stress at 65% strain and absorbed energy of the composites with 15% milled glass fiber increases by 131.7% and 159.9%, respectively, while those of pure polyurea increases only 99.5% and 121.0%, respectively. The fiber length affects the properties of the composites in the same manner. However, its effect is smaller than that of fiber volume fraction. As the fiber length increases in the range studied, the increments of the stress at 65% strain and absorbed energy are lower than 64.3% and 127.2%, respectively. Furthermore, Figure 6 shows the stress at 10% strain of the composites at strain rates in the range from 0.001 to ~3000 s^−1^. It is worth noting that the strain rate sensitivity of the MG_f_/PU composites is much more significant when the strain rate is higher than 1 s^−1^.

These results are easy to understand. In this study, a silane-based surface treatment was applied to the surface of the glass fiber used, and as a result a strong interfacial bonding between the fiber and the polyurea matrix was achieved. When a load is exerted on the composites, it can be effectively transferred to the fiber through the fiber–matrix interface, and the fiber acts as a load-bearing component. The higher the fiber content, the more load it bears, resulting in a more obvious strengthening effect. In addition, the load-bearing capacity of short fiber is related to its length. The stress on the fiber increases gradually from the end of the fiber to the middle of the fiber, until the stress at the end of the fiber reaches the yield strength of the matrix. Hence, the load-bearing capacity of the fiber increases as the length of the fiber increases. Furthermore, the fibers implanted occupy the space between polyurea molecular chains and restrict their movement, leading to an improved load-bearing capacity. Apparently, such an effect increases with increasing fiber content and fiber length.

### 3.4. Dynamic Mechanical Properties

Figure 7 shows the DMA storage (*E*′) and loss (*E*″) moduli as a function of temperature for the MG_f_/PU composites with varying fiber volume fractions and lengths at 1 Hz. Some detailed data are given in Table 2.

Similar to the behavior of pure polyurea, a precipitous decline in the *E′* and *E″* of the composites is observed. The *E′* and *E″* of the composites are generally improved by the incorporation of milled glass fiber and strongly depend on the fiber volume fraction. As shown in Table 2, *E*′ and *E*″ of the composites at −20 and 50 °C both increase monotonically with fiber content. Compared with that of pure polyurea, *E*′ and *E*″ of the L15/PU composites increase by over 95% and 75% at −20 °C and increase over 121% and 113% at 50 °C, respectively. Since *E*′ is proportional to the energy stored elastically, and *E*″ refers to the energy transferred to heat irreversibly, it could be concluded that the addition of glass fibers into polyurea could not only improve the stiffness of the composites but also increase its energy dissipative capability. The more important role played by the fiber in *E*′ of the composites at higher temperatures may be due to the increased contrast between the stiffness of the fiber and matrix. The increase in *E*″ may be explained by two reasons: First, the degree of hard segment mixing in the soft phase increases, which is in excellent agreement with the FTIR observations. Second, the chain relaxation dynamics of the polyurea matrix may slow down due to the interaction between the milled glass fibers and the polyurea matrix [44,45].

A similar increase in *E*′ and *E*″ of the MG_f_/PU composites because of the increasing fiber length can also be observed. The *E*′ spectra of the S10/PU composites is close to that of pure polyurea, while *E*′ and *E*″ of the L10/PU composites increase by over 89% and 74% at −20 °C and increase over 97% and 129% at 50 °C, respectively. *E*′ is an important property to assess the load-bearing capability of a composite material [46], and as mentioned above, the load-bearing capacity of short fiber increases with the increase in length. Thus, the *E*′ of the MG_f_/PU composites increases. *E*″, which indicates the energy dissipative capability, refers to the viscous nature of the material. As the milled glass fibers are introduced, the longer the fiber, the slower the flow and the higher the resulting *E*″.

## 4. Conclusions

In this study, milled glass fiber with various length was introduced into polyurea, and two material series were prepared to study the effects of fiber volume fraction and fiber length on the morphological and mechanical properties of the MG_f_/PU composites, such as compressive properties at different strain rate and dynamic mechanical properties as a function of temperature. The results suggested that the milled glass fiber is distributed homogenously and randomly in the polyurea matrix, and no significant sign of agglomerates is observed. The interfacial adhesion between the fibers and the polyurea matrix is good, especially for the composites with longer fiber. Moreover, it was found that the H-bond strength of polyurea was not altered by the presence of milled glass fibers; however, the incorporation of milled glass fiber leads to more phase mixing and decreased hydrogen bonding. The compressive stress–strain behavior of MG_f_/PU composites is highly nonlinear and strongly rate-dependent. The strain rate sensitivity of the composites increases with the fiber volume fraction and fiber length. Furthermore, the mechanical properties of the composites are significantly affected by the fiber volume fraction and mean fiber length. The elastic modulus, stress at 65% strain, energy absorption capability and the storage and loss moduli all increase with fiber volume fraction and fiber length. However, the effect of the fiber content is more predominant. It is worth noting that the introduction of milled glass fiber improves both the stiffness and the energy dissipative capability of the MG_f_/PU composites, which is promising for the application of these materials.

## Figures and Tables

**Figure 1 polymers-14-03080-f001:**
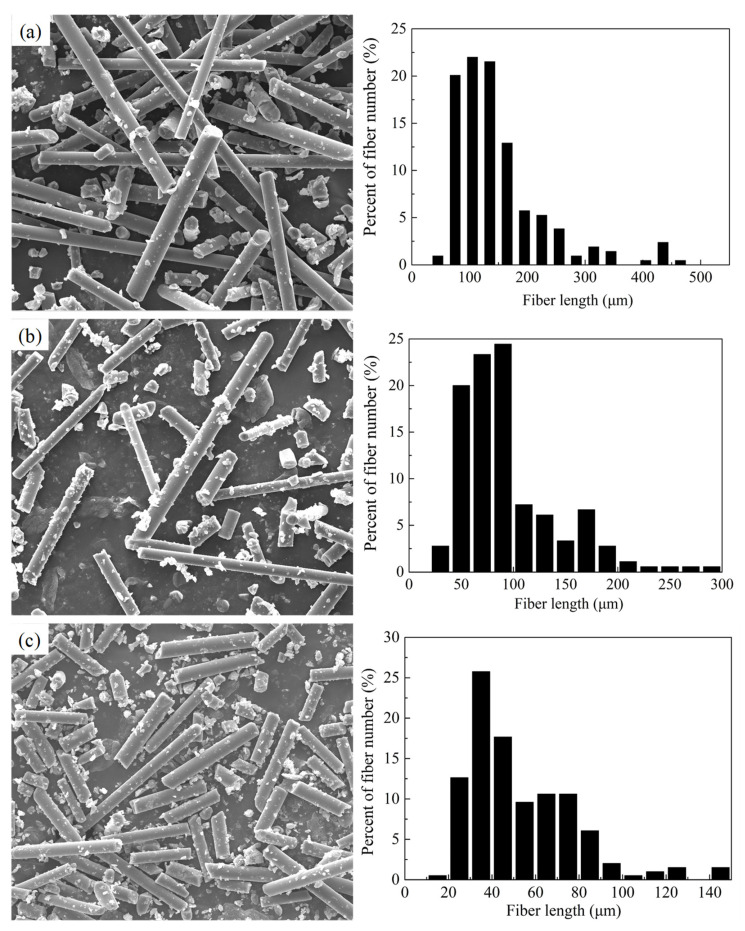
Morphology of milled glass fibers and corresponding fiber length distributions. (**a**) 20 mesh, (**b**) 50 mesh, (**c**) 200 mesh.

**Figure 2 polymers-14-03080-f002:**
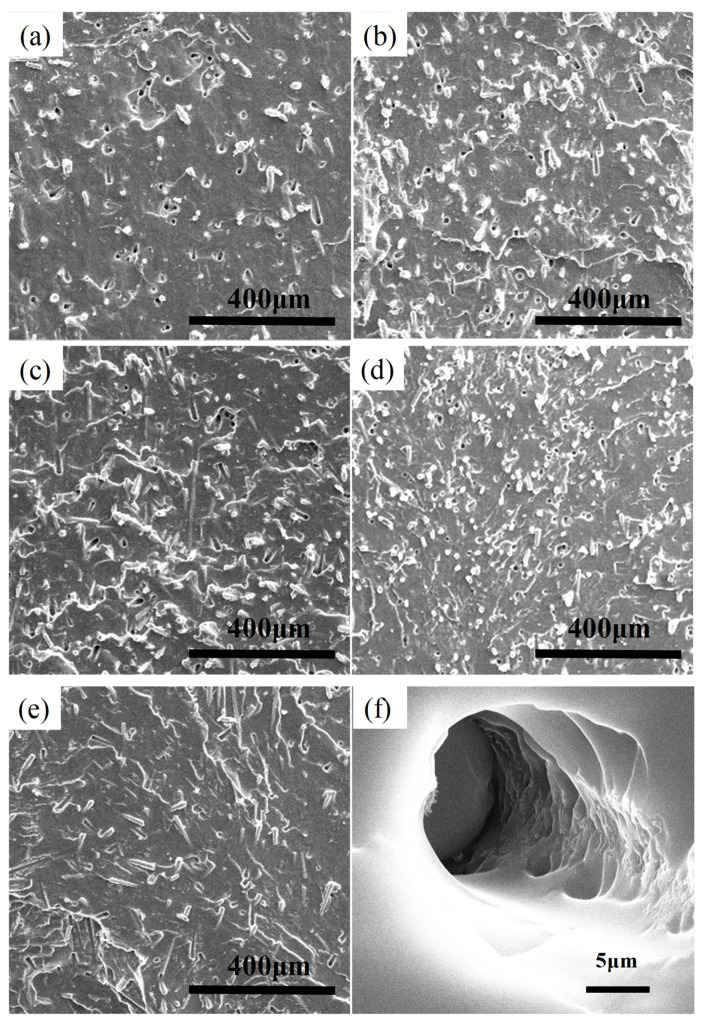
Morphology of MG_f_/PU composites. (**a**) L5/PU, (**b**) L10/PU, (**c**) L15/PU, (**d**) S10/PU, (**e**) M10/PU, (**f**) magnification of a hollow cylinder in Figure 2a.

**Figure 3 polymers-14-03080-f003:**
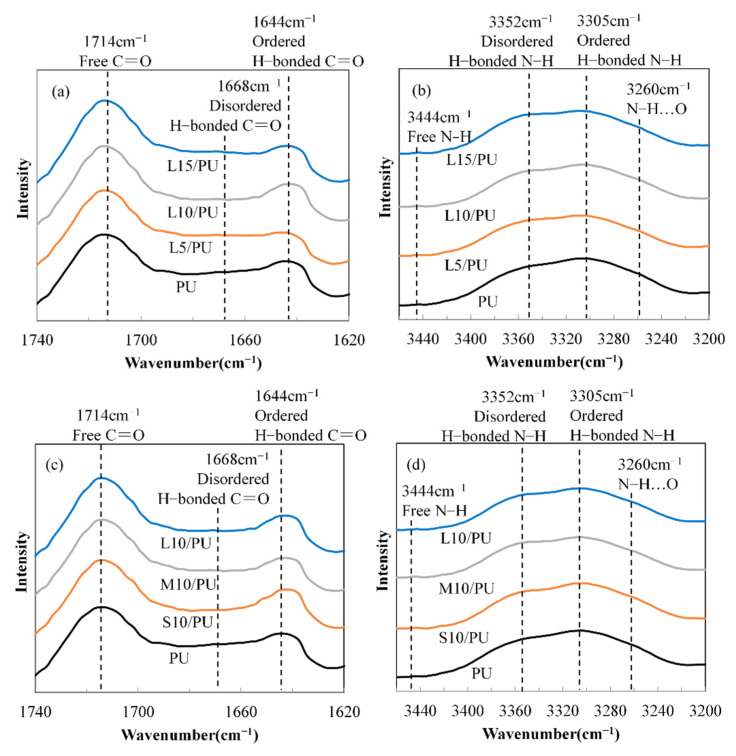
FTIR spectra of MG_f_/PU composites with varying fiber volume fractions (**a**,**b**) and fiber lengths (**c**,**d**) in the C=O (~1740–1620 cm^−1^) and N-H stretching regions (~3460–3200 cm^−1^).

**Figure 4 polymers-14-03080-f004:**
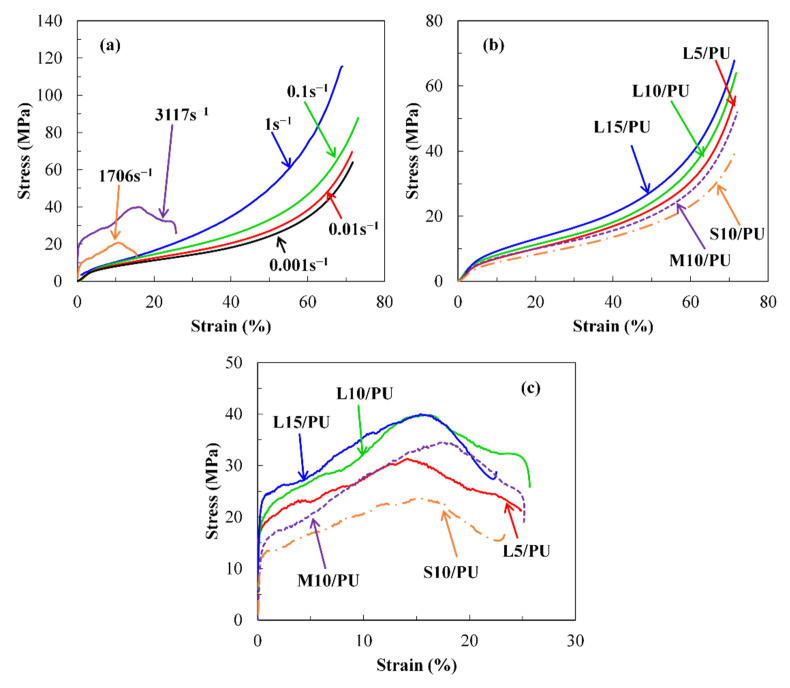
Compressive stress–strain curves of MG_f_/PU composites. (**a**) L10/PU composites with varying strain rates, (**b**) MG_f_/PU composites with varying fiber lengths and volume fractions at the strain rate of 0.001 s^−1^, (**c**) MG_f_/PU composites with varying fiber lengths and volume fractions at the strain rate of ~3000 s^−1^.

**Figure 5 polymers-14-03080-f005:**
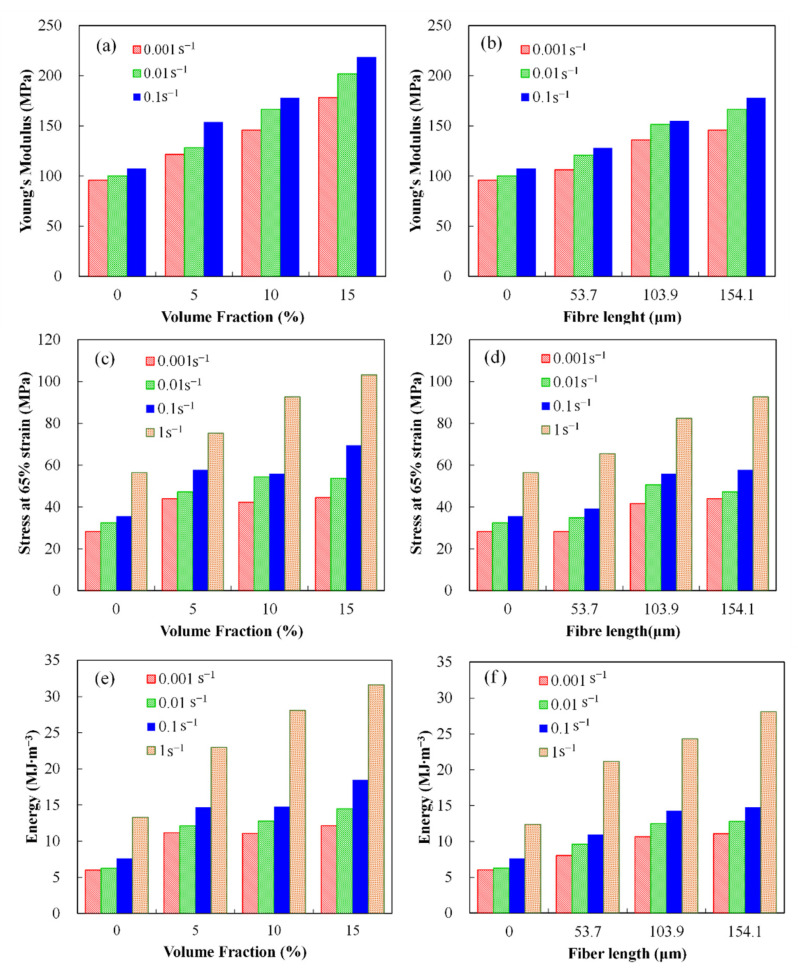
Compressive mechanical properties of MG_f_/PU composites as a function of fiber volume fraction (**a**,**c**,**e**) and fiber length (**b**,**d**,**f**) at strain rates in the range from 0.001 to 1 s^−1^: (**a**,**b**) Young’s modulus, (**c**,**d**) stress at 65% strain, (**e**,**f**) absorbed energy.

**Figure 6 polymers-14-03080-f006:**
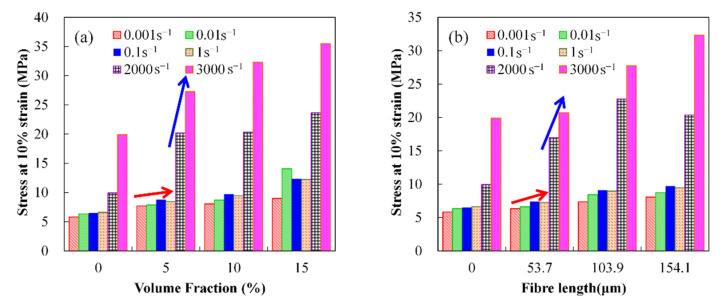
Stress of 10% strain of MG_f_/PU composites as a function of fiber volume fraction and fiber length at the strain rates of ~3000 s^−1^.

**Figure 7 polymers-14-03080-f007:**
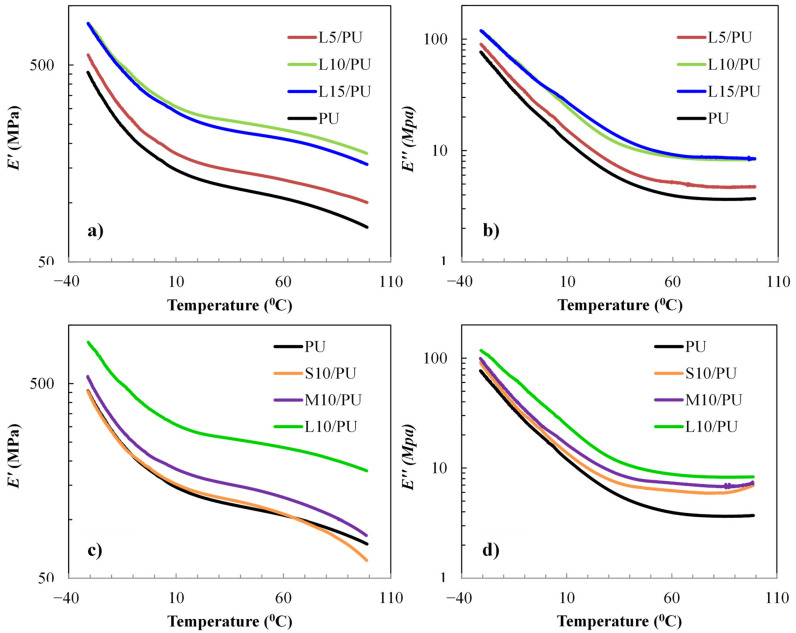
Effect of fiber volume fraction (**a**,**b**) and fiber length (**c**,**d**) on dynamic mechanical properties of various MG_f_/PU composites as a function of temperature at 1 Hz.

**Table 1 polymers-14-03080-t001:** The carbonyl hydrogen bonding index, the degree of phase separation (*DPS*) and the degree of phase mixing (*DPM*) in pure polyurea and MG_f_/PU composites.

	*R*	*DPS*	*DPM*
PU	1.254	55.6%	44.4%
L5/PU	1.019	50.5%	49.5%
L10/PU	0.996	49.9%	50.1%
L15/PU	0.977	49.4%	50.6%
S10/PU	1.130	53.0%	47.0%
M10/PU	1.006	50.2%	49.8%

**Table 2 polymers-14-03080-t002:** Dynamic mechanical properties of MG_f_/PU composites at −20 and 50 °C.

	−20 °C		50 °C	
	*E*′ (MPa)	*E*″ (MPa)	*E*′ (MPa)	*E*″ (MPa)
PU	288.6	43.9	111.1	4.4
L5/PU	353.9	53.0	137.2	5.5
L10/PU	546.9	76.5	219.6	10.1
L15/PU	562.8	77.2	245.6	9.4
S10/PU	283.9	48.7	115.6	6.5
M10/PU	336.7	54.0	139.5	7.6

## Data Availability

The data that support the findings of this study are available from the corresponding author (Jing Qiao) on request.

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
