# Peer review of "Effects of Fiber Volume Fraction and Length on the Mechanical Properties of Milled Glass Fiber/Polyurea Composites"

_polymers, 2022, doi:10.3390/polym14153080_

Round 1

Reviewer 1 Report

The work deals with fabricating a polyurea composite reinforced with milled glass fiber. The authors studied the effect of length and volume fraction of milled glass fiber in the polyurea matrix on its mechanical properties to optimise the fiber density and length. The manuscript is well designed and supported with results. It can be published after minor corrections below:

C1) Please check the abbreviation of Fourier transform infrared spectroscopy within the whole manuscript. It is supposed to be FTIR, however, many “FITR” are seen in the text. 

C2) The reason for choosing stress at 65% strain needs to be explained within the manuscript. 

C3) Line 301: Figure 5 a to f needs to be explained in the figure caption, too.

Reviewer 2 Report

Recommendation: Accept in present form.

Comments:

The paper by Qiao et al. contributes a review of research efforts on polyurea (PU) reinforced with milled glass fiber (MGf). The title and abstract are appropriate for the content of the text.  The article gives an interesting historical and scientific perspective on polyurea characterization. It would be easier to position this work and provide a clear view if the author could introduce these works in an orderly sequence. The addition of milled glass fiber into polyurea not only improves the stiffness of the composites but also increases its energy dissipative capability.
